# Parking CAR T Cells in Tumours: Oncolytic Viruses as Valets or Vandals?

**DOI:** 10.3390/cancers13051106

**Published:** 2021-03-05

**Authors:** Laura Evgin, Richard G. Vile

**Affiliations:** 1Canada’s Michael Smith Genome Sciences Centre, BC Cancer, Vancouver, BC V5Z 1L3, Canada; levgin@bcgsc.ca; 2Department of Medical Genetics, University of British Columbia, Vancouver, BC V6H 3N1, Canada; 3Department of Molecular Medicine, Mayo Clinic, Rochester, MN 55905, USA; 4Department of Immunology, Mayo Clinic, Rochester, MN 55905, USA

**Keywords:** oncolytic virus, adoptive T cell therapy, CAR T cell, immunotherapy

## Abstract

**Simple Summary:**

Chimeric Antigen Receptor (CAR) modified T cell therapy has revolutionized the treatment of B cell malignancies, however transposition of the technology to the solid tumour setting has been met with more therapeutic resistance. Oncolytic Viruses (OVs) are multi-modal agents, possessing tumour cell cytolytic capabilities as well as strong immune stimulatory properties. Although combination therapy poses great promise, great care must be employed so as to maximize the output of each modality and minimize interference.

**Abstract:**

Oncolytic viruses (OVs) and adoptive T cell therapy (ACT) each possess direct tumour cytolytic capabilities, and their combination potentially seems like a match made in heaven to complement the strengths and weakness of each modality. While providing strong innate immune stimulation that can mobilize adaptive responses, the magnitude of anti-tumour T cell priming induced by OVs is often modest. Chimeric antigen receptor (CAR) modified T cells bypass conventional T cell education through introduction of a synthetic receptor; however, realization of their full therapeutic properties can be stunted by the heavily immune-suppressive nature of the tumour microenvironment (TME). Oncolytic viruses have thus been seen as a natural ally to overcome immunosuppressive mechanisms in the TME which limit CAR T cell infiltration and functionality. Engineering has further endowed viruses with the ability to express transgenes in situ to relieve T cell tumour-intrinsic resistance mechanisms and decorate the tumour with antigen to overcome antigen heterogeneity or loss. Despite this helpful remodeling of the tumour microenvironment, it has simultaneously become clear that not all virus induced effects are favourable for CAR T, begging the question whether viruses act as valets ushering CAR T into their active site, or vandals which cause chaos leading to both tumour and T cell death. Herein, we summarize recent studies combining these two therapeutic modalities and seek to place them within the broader context of viral T cell immunology which will help to overcome the current limitations of effective CAR T therapy to make the most of combinatorial strategies.

## 1. CARs: The Ultimate Tumour Killing Machines?

An insufficiency of the breadth, or functionality, of the tumour reactive T cell repertoire can be overcome through the use of adoptive T cell therapy (ACT) in which T cells specific to the antigenic constituency of the tumour are generated ex vivo and re-introduced to the patient. These cells may be directly expanded from the tumour [1] or derived from peripheral blood in which novel specificity is conferred by expression of an ectopic T cell receptor (TCR) or a chimeric antigen receptor (CAR) [2]. This allows for T cells to be cultured to possess desirable phenotypes in vitro, and patients to be treated with preconditioning regimes to promote T cell engraftment and minimize suppression. T cells generated through these means have led to a large subset of complete responses in otherwise treatment refractory disease, with CAR T cells in particular experiencing unprecedented success against B lymphoid cancers [3].

While TCRs recognize intracellular antigens presented in the context of the major histocompatibility complex (MHC), the synthetic CAR confers specificity in an MHC unrestricted manner to cell surface, and now recently, soluble antigens [4]. The CAR design is modular with each domain contributing to the resulting functional outcome. In its most basic configuration, the CAR molecule is composed of an extracellular antigen binding domain (most commonly an scFv), an extracellular hinge region, a transmembrane domain, and an intracellular signaling domain (including the CD3ζ and costimulatory domains) [5].

The signaling region of the earliest iterations of CARs comprised only the CD3ζ endodomain and triggered effector function but had limited therapeutic potential [6,7]. A multi-step activation model is required to mount effective T cell responses, with signal one being derived from the TCR, signal two from co-stimulatory ligation, and signal three from cytokine exposure. Co-stimulatory domains have thus been included proximal to CD3ζ in second generation constructs to promote persistence and anti-tumour activity [8,9,10]. Third and fourth generation CARs include two or more co-stimulatory domains, or other transgenes including cytokines [11,12,13,14]. CD28 and CD137 (4-1BB) have been most rigorously explored preclinically and are included in the clinically approved constructs axicabtagene ciloleucel (Yescarta), and tisagenlecleucel (Kymriah), respectively [15,16,17]. Several other costimulatory domains have however been successfully evaluated including CD27, OX40, CD40L, and ICOS, and the incorporation of distinct costimulatory domains has been shown to have profound effects on phenotype, expansion kinetics, metabolism, and persistence [18,19,20,21,22]. Sophisticated synthetic biology circuits have been designed to incorporate additional regulatory tuning and recognition capacity [23]. The functional properties of each domain have been explored and are extensively reviewed elsewhere [24,25,26,27].

Retro- or lentiviral vectors are the primary means of introducing stable expression of the CAR into T cells, but random integration into the genome has the potential to lead to insertional mutagenesis and variegated expression of the CAR, thus prompting use of targeted means to introduce the CAR into genomic safe harbours such as the T-cell receptor α constant (TRAC) locus using CRISPR/Cas9 [28,29]. Although the CAR has primarily been introduced into autologous T cells to generate a bespoke patient product, disruption of the TRAC simultaneously enables the use of allogeneic T cells by preventing the development of graft vs. host disease (GVHD) [30].

## 2. Switching on the Ignition with Oncolytic Viruses

Through natural tropism, or genetic engineering, oncolytic viruses are a class of viruses which share a preference for replication in malignant cells over normal tissue. A broad diversity of viruses has been defined as having oncolytic properties and possess distinct genomes (RNA and DNA), entry specificities, replication mechanics, immune-evasion machinery, and genetic modifications which collectively confer tumour specificity. Infection of tumour cells can be facilitated by the overexpression of viral binding and entry receptors, such as CD46 for measles [31] or ICAM-1 for Coxsackievirus A21 [32]. Deletion of viral genes, complemented by high level expression in tumour cells, such as those involved in nucleotide metabolism (thymidine kinase, ribonucleotide reductase, uracil DNA glycosylase) has been employed with herpes simplex virus (HSV) and vaccinia virus (VACV) strains [33,34,35,36]. The adenovirus E1A protein binds the cellular retinoblastoma protein to drive S-phase entry, allowing it to access the cellular DNA replication and protein synthesis machinery, and a 24 amino acid deletion restricts the virus to rapidly proliferating cells [37].

The primary innate antiviral mechanism, type I interferon (IFN), is known to be anti-angiogenic, and to promote growth arrest and apoptosis [38]. While many tumours harbour mutations in key IFN genes or epigenetically silence them, the activation of oncogenic pathways or loss of tumour suppressors such as EGFR, Wnt B catenin, or Pten, all have links to IFN production or responsiveness [39,40]. The net effect is that an estimated 65–70% of cancer cell lines are thought to have defects in their ability to produce, or respond to, type I IFN [41]. Compromised IFN signaling thus underlies the tumour selectivity of OVs and is particularly relevant to Vesicular stomatitis virus (VSV) and Newcastle disease virus (NDV) [41,42]. The safety and specificity of oncolytic VSV is further enhanced through deletion or mutation at position 51 in the matrix protein whose normal function is to block nucleocytoplasmic trafficking of mRNA, thus preventing the translation of IFN and interferon stimulated genes (ISGs) [41,43]. An analogous approach encodes IFNβ in the viral genome, with the added benefit that the cytokine promotes dendritic cell (DC) activation and acts as a signal 3 cytokine for T cell priming [44,45,46].

## 3. Combination OV and T Cell Therapies: Driving CAR T to the Tumour

Inter and intra-patient tumour heterogeneity, the plastic nature of cancer genomes, and the dynamic state of the tumour microenvironment all contribute to the likely failure of monotherapy approaches to cancer treatment. In this respect, it would seem on the surface that a partnership between OVs and CAR T cells offers a perfect opportunity to orchestrate a multi-pronged approach against often rapidly evolving targets on multiple fronts. Although combination strategies using multiple biologic agents may face more regulatory hurdles, significant clinical development of both platforms individually may pave a way forward. Significant toxicities have been well described for CAR T therapy, including cytokine release syndrome (CRS) and neurotoxicity, and it will be paramount to establish a robust safety profile of any combination strategy. A multiplexed approach has now made the jump from the bench [47] to clinic as the investigation of HER2 CAR T cells and oncolytic and helper dependent adenovirus expressing IL12 and anti-PDL1 is now underway (NCT03740256).

Herein we review how intrinsic and engineered properties of oncolytic viral vectors may be exploited to enable CAR T to overcome barriers to effective therapy in the solid tumour setting, including restricted infiltration, interaction with immunosuppressive soluble mediators and cellular players, and antigen heterogeneity and escape (Figure 1). However, combination with OVs does not automatically guarantee a superior therapeutic outcome as they can lead to both helpful and deleterious consequences for CAR T cells, and thus act both as valets and vandals. The studies highlighted herein illustrate the complex biology of each living drug and the importance of highly tailored therapeutic strategies.

## 4. The TME—A Breaker’s Yard for CAR T Cells

The tumour microenvironment is composed not only of cancer cells, but heterogenous levels of a variety of immune cell types whose location and density can profoundly affect prognosis and therapeutic response [48]. Although effector T cells, NK cells and B cells can be present to variable degrees, suppressive immune cell types, including regulatory T cells (T_reg_) and aberrantly matured myeloid cells such as myeloid derived suppressor cells (MDSCs) and tumour-associated macrophages (TAMs) are often found within the tumour core and the invasive front [48,49]. This constellation of stromal cells coordinates a network of overlapping regulatory mechanisms which mask the tumour from immune destruction, beginning with the expression of chemokines which disfavour the recruitment of effector T cells. Expression of the counter-ligands for CXCR3 on activated lymphocytes, CXCL9, 10, and 11, is associated with good prognosis and is required for T-cell trafficking across tumour vascular checkpoints [50,51,52]. However, tumours may reduce levels of these ligands through epigenetic silencing or the co-expression of chemokine- cleaving proteases [53,54], and instead express CCL2 which recruits immature myeloid cells and TAMS [55,56,57]. In turn, MDSCs and TAMs lead to metabolic and cytokine mediated dysfunction of T cells through the production of arginase 1, inducible NO synthase (iNOS), indoleamine 2,3-dioxygenase (IDO), transforming growth factor beta (TGFβ) and IL10, respectively [58,59]. T_regs_ exploit cytokine-mediated and contact-dependent mechanisms to limit effector T function, including competition for IL2, expression of CD39 and CD73 leading to the production of adenosine, secretion of TGFβ and IL10, and expression of checkpoint ligands such as CTLA-4 and PDL-1 [60]. Finally, tumour cells themselves, immune cells, and exosomes can all express ligands or release soluble factors which engage checkpoint receptors on CAR T cells, including PD-1, TIM-3, LAG-3 and TIGIT, leading to dysfunction and apoptosis [61]. Overall, these factors contribute to making the tumour more akin to a scrap yard than the exclusive valet parked ramp where you would, ideally, want to leave your meticulously engineered CAR.

## 5. TME Make over by Oncolytic Viruses

OV infection leads to a cascade of inflammatory events, stimulating innate and adaptive immune responses, and thus changing the cytokine, chemokine and cellular composition of tumours. Viral nucleic acids serve as pathogen-associated molecular patterns (PAMPs) to activate cytoplasmic RNA and DNA sensors and Toll-like receptors (TLRs), converging on TRIF and MyD88 to activate type I IFN signaling [62,63]. OV infection upregulates calreticulin (CRT) on the cell surface, and oncolysis releases adenosine triphosphate (ATP) and high-mobility group box 1 (HMGB1) into the extracellular environment, all members of the danger-associated molecular pattern (DAMP) family [64,65]. Together, in concert with type I IFN, these signals promote the recruitment and maturation of DCs which take up virus and tumour debris, traffic antigen back to lymph nodes, and prime naïve T cells. Although all CXCR3 ligands are induced by IFNγ, CXCL10 and 11 are directly agonized by type I IFN [66].

Thus, infection leads to a global change in the cellular composition of the tumour and the corresponding derived soluble mediators. Mouse models have demonstrated that OVs promote the infiltration and activation of CD8 T cells, CD11c+ DC, NK cells, M1-like macrophages, and concomitantly reduce the proportion of Tregs and MDSCs [67,68,69,70,71,72,73,74]. Although a large fraction of infiltrating T cells is likely to be specific to viral antigens, oncolysis can act as a tumour antigen agnostic vaccine, priming T cells against public and private neoantigens [73,75]. Oncolytic infection and type I IFN concomitantly induce the upregulation of checkpoint receptor ligands such as PDL1, and thus combination therapy with pharmacologic or viral expression of checkpoint blocking antibodies with vaccinia [67], VSV [68,76], reovirus [77], measles [70,78], HSV [72] and NDV [71] have provided superior tumour outcomes. Immune correlative studies in the clinical setting have corroborated preclinical findings, showing that talimogene laherparepvec (HSV) and reovirus treatment promotes an increase in CD8 T cell density in post treatment biopsies [79,80], and measles treatment facilitates T cell priming against tumour antigens [81]. The representation of virus specific or tumour antigen specific T cells which infiltrate into a tumour is not well characterized, however is likely to be skewed toward viral specificities due to high level expression of viral epitopes which are not subject to tolerance mechanisms.

## 6. Fiddling while CARs Burn

On the face of it, these pro-inflammatory effects of OV infection of tumours would be predicted to provide an excellent make over to convert the tumour from a parking site that is essentially ‘closed for business’ to T cells to one that now is replete with special offers for both long and short term parking deals -suggesting that OVs should significantly improve CAR T efficacy in the solid tumour setting. However, woven throughout the potential benefits of type I IFN responses on T cell recruitment/activation are negative feedback mechanisms which have evolved to promote de novo anti-viral T cell responses, and subsequently restrain inflammation to prevent autoimmunity. While the upregulation of checkpoint ligands can be blocked by the simultaneous use of checkpoint inhibitors, the more insidious role of type I IFN is its direct effect on CD8 T cell biology. Depending on the timing, memory status and concentration, type I IFN exerts pleiotropic effects on T cells. Type I IFN supports the expansion and differentiation of naïve T cells, thus playing a key role in cell fate decisions as a signal 3 cytokine [46,82]. However, via a mechanism that is thought to make space for T cells specific to incoming pathogens, type I IFNs also promote acute apoptosis of memory T cells [83,84,85,86]. Indeed, priming the tumor with oncolytic VSV expressing IFNβ simultaneously promoted significant CAR T attrition in a type I IFN dependent manner [87]. Although the effect was largely T cell intrinsic as adoptively transferred transgenic Pmel T cells underwent the same IFN associated fate, additional CAR specific effects were observed. Virus derived IFN upregulated the expression of the CAR, promoting downstream effects of tonic signaling, including high level expression of inhibitory receptors. Although apoptosis was averted through the use of transgenic or CRISPR edited interferon alpha receptor (IFNAR1) deficient T cells, and thus allowed for enhanced combination therapy in lymphodepleted animals, this engineering strategy inadvertently sensitized the CAR T cells to NK cell attack [87,88,89]. These effects are thought to be broadly relevant to other OVs. While the underlying biology was enhanced by the expression of IFNβ from the VSV vector, oncolytic reovirus also induced CAR T cell attrition, albeit to a more moderate extent [87].

High levels of VEGF in the tumor have been shown to attenuate type I IFN signaling in tumour-associated endothelial cells through Blimp-1, thus sensitizing them to OV infection [90]. Together with the neutrophil-dependent induction of microclots, several OVs, including reovirus, VSV, vaccinia virus, and NDV, have been shown to induce vascular shutdown in tumours [90,91,92,93,94,95]. While vascular collapse may starve tumour cells, it may simultaneously limit the access of CAR T cells to their targets. Vascular normalization using 3TSR prior to NDV therapy has been shown to increase immune cell trafficking into the tumour [95], and may thus represent an important third-party consideration for combination therapy.

## 7. Viruses as Micro-Pharmacies for T Cells

Although the virus-intrinsic effects of infection on the tumour composition are potentially overwhelmingly favourable, an additional therapeutic strength may be in the ability of OVs to deliver desirable transgenes locoregionally. The magnitude and timing of chemokine induction varies depending on virus biology and several OVs have been engineered to express chemokines to enhance recruitment of CAR T to the tumour. Oncolytic adenovirus armed with the chemokine RANTES (CCL5) to promote infiltration, as well as the cytokine IL15 to support T cell survival once in the tumour conferred enhanced therapeutic benefit when used in combination with GD2 CAR T [96]. A similar strategy incorporating the CXCL11 transgene into vaccinia virus enhanced CD8 T cell infiltration and enhanced mesothelin specific CAR T therapy of murine TC1 tumours [97].

In order to sidestep any reduction in replication and oncolytic capacity, Shaw et al. used a gutted (helper-dependent) adenovirus to deliver various cytokine payloads in combination with replication competent oncolytic adenovirus and HER2 specific CAR T. Among the candidate cytokines IL2, IL7, IL-12p70, IL15, and IL2, expression of IL-12p70 was found to potentiate CAR T efficacy in a xenograft model of head and neck squamous cell carcinoma [47]. Further incorporation of a PD-L1 blocking antibody in the helper dependent adenovirus increased anti-tumour efficacy [47,98] and provides the rationale for clinical evaluation. Notably, local production of the anti-PDL1 antibody from the virus was superior to systemic administration of anti-PD-L1 IgG, thus highlighting the benefit of in situ transgene production [98]. Encoding the checkpoint blockade molecule and IL12 in the virus is a particularly attractive strategy to produce locally high concentrations at bioactive sites, where systemic delivery is associated with adverse events [99].

A similar strategy by Wanatabe et al. employed oncolytic adenovirus armed with TNFα and IL2 to enhance both human and mouse mesothelin specific CAR T. Treatment with Ad5/3-OAd-TNFα-IL2 induced more robust and persistent localization of human CAR T in the tumour and correspondingly induced sustained regression. Adenovirus encoding the murine cytokines increased CD80 and CD86 expression on tumour resident macrophages and dendritic cells, upregulated CXCL10 production in the tumour, and was associated with higher levels of infiltrating CD4 CAR T, and CD4 and CD8 endogenous T cells; all of which contributed to increased tumour control in the combination arm [100].

## 8. Graffitiing Antigenic Specificity onto Tumours

Oncolytic viruses exhibit a range of infectivity within tumours suggesting that highly susceptible cells would be killed, whilst cells refractory to oncolysis, but in which viral genes are expressed, could be targeted by CAR T. In solid tumours where no specific CAR targets have been identified, OVs could be used to deliver ectopic antigens to tumour cells [101,102]. This strategy could also be applied to re-target CAR T cells to antigen negative tumour cells, a common mechanism of treatment failure [103,104]. Proof of concept studies from Aalipour et al. and Park et al. used oncolytic vaccinia viruses to decorate cancer cells with CD19 and demonstrated targeting of CD19 CAR T to previously unrecognized tumour cells both in vitro and in vivo. Although CD19 delivery leads to unnecessary induction of B cell aplasia, the modular nature of the approach suggests that it could be extended to other antigens. One significant limitation would be heterogenous expression of the CAR target antigens and incomplete targeting by either modality. However, both OVs and CAR T have been demonstrated to elicit endogenous T and B cell responses to tumour associated antigens (TAA) through epitope spreading [80,105,106,107], and indeed Park et al. show that mice which were cured by combination therapy were partially protected against re-challenge with parental tumours which did not express the CAR antigen.

The immune adjuvanticity and transgene expression capabilities of OVs allow them to act as strong vaccines. TAAs can be expressed to a high degree in tumour cells, and upon cell lysis, taken up by dendritic cells or other phagocytic cells for presentation to T cells. Viruses may also initiate abortive non-lytic infections in non-transformed cells, such as APCs, which lead to the expression of viral genes and the subsequent priming of T cells. Thus, oncolytic vaccination against single TAAs, or even a library of antigens, provides stronger anti-tumour therapy than parental strains [108,109,110]. The size of the T cell pool is further magnified through the use of heterologous vectors for priming and boosting which encode the same antigen [111,112,113], or by adoptive transfer of transgenic antigen specific T cells [114,115,116]. Although CAR modified T cells acquire a novel specificity, they also retain the specificity conferred by their native TCR, and parallel work has sought to boost CAR T cells through the TCR. Transgenic T cells expressing both a CAR specific to HER2 and bearing a TCR specific for either the gp100 or OVA antigens are expanded by treatment with vaccinia expressing the cognate epitope, leading to accumulation of T cells in the tumour and eradication of large established tumours [117]. Clinical evaluation of a mixed infusion product containing Epstein-Barr Virus (EBV) TCR specific and open TCR repertoire GD2 CAR T demonstrated that virus specific CAR T circulate at a higher frequency than the counterpart control T cells [7]. Viral reactivation or vaccination further supports the expansion of virus specific CAR T cells [118,119].

Bispecific T-cell engager (BiTE) technology links an anti-CD3 scFv to a tumour antigen specific scFv, and thus bypasses both the TCR-MHC interaction and the CAR to engage effector function. In this way, OVs and CAR T cells engineered to express BiTEs can redirect endogenous T cells, or CAR T themselves, against a second tumour antigen specificity [120,121,122,123]. Putting these platforms together, Wing et al. demonstrated that oncolytic adenovirus expressing an EGFR-targeting BiTE improved the activation, proliferation, and cytokine production of CART cells targeting the folate receptor alpha (FR-α) and can help to overcome antigen heterogeneity [124]. Use of the virally expressed BiTE redirected both CAR T and CAR negative nonspecific T cells and provided superior anti-tumour efficacy compared to each monotherapy. A multiplex strategy has also combined BiTE expression (specific to CD44v6) with cytokine (IL12) and checkpoint (anti-PDL1) delivery using the oncolytic and helper dependent adenovirus system in conjunction with HER2 CAR T to combat several mechanisms of tumour escape simultaneously [125].

## 9. Virus CAR-Pooling to Tumours

Although we have discussed primarily the use of viruses to improve various aspects of T cell anti-tumour function, so T cells can also be exploited to improve the efficiency of oncolytic virotherapy—most notably perhaps through helping to deliver the viruses to their active site. In vitro pre-loaded antigen specific T cells, and cytokine-induced killer cells, have been reported to traffic OVs, allowing replication and oncolysis within the tumour. This smuggling of viruses to tumours has been shown to be possible even in pre-vaccinated hosts, thus bypassing circulating anti-viral antibodies which have often proved to be the Achilles heel of systemic OV therapy [126,127,128,129,130,131]. Similarly, murine and human HER2 CAR T cells loaded with low doses of oncolytic VSV or vaccinia virus have been shown to deposit their cargo without compromising the function of the CAR T cells [132]. As discussed above, the TME represents a very unwelcoming parking place for (CAR) T cells. The same is true for highly immunogenic viruses trying to passage through and then exit selectively from a circulatory highway system heavily patrolled by neutralising antibodies, complement, and other anti-viral effectors. However, CAR-pooling of precisely engineered tumour-targeting viruses may overcome several of the barriers to effective combination CAR/OV therapy.

## 10. Conclusions: OV Enhanced CAR T Cell Therapy—And Vice Versa

In the quest for a systemic, potent anti-tumour therapy, both adoptive (CAR) T cell transfer and oncolytic viruses have enormous curative potential. Both intrinsic and engineered capabilities endow OVs with a unique potential to serve as a platform to enhance adoptive T cell therapy. However, in order to reach their destination, both (CAR) T cells and viruses have to navigate a circulatory highway fraught with diversions, patrols and obstacles. Even if/when they successfully reach the tumour, the TME represents a highly immune-suppressive, neutralizing destination. This neighbourhood is unlikely to appeal to any owner of such highly sophisticated and engineered anti-tumour killing machines (cells or viruses) as a safe and effective parking place. However, tumour infection by OVs has the potential to effect a dramatic make over and convert this hostile, T cell repellent TME into a highly attractive haven, open for business for an influx of CAR T cells. In this respect, the inflammatory profile induced by OV infection, as well as OV-triggered transgene expression, needs to be carefully crafted. Moving forward, models in which the safety and efficacy of combination strategies that intricately engage innate and adaptive immunity are evaluated must account for factors which both recruit and support activated CAR T cell therapies, as well as those compensatory mechanisms which restrain and inhibit the (T cell) immune system. Importantly, a variety of models should be used which, combinatorially, analyze the plethora of factors which may be absent from specific model systems, or which may be present but which are non-reactive in the specific model being tested—such as type I IFN, which binds to species specific IFNAR [133]. Thus, it will be critical to prevent OV infection from simply converting a T cell freezing TME into an incendiary, CAR T-vandalizing, TME. With appropriate design of the levels, nature and timing of inflammatory cytokine expression from OV infection, it will be possible to generate an optimal, climate-controlled environment that nurtures the gentle, valet parking of CAR T cells inside the tumour—where the T cells can go on to do their worst against solid tumours. Therefore, we believe that by generating novel designer combinations of paired viruses and engineered T cells it will be possible to create a powerful synergy between adoptive T cell therapies and OV infection, whereby each one enhances the tumour trafficking, selectivity and potency of the other.

## Figures and Tables

**Figure 1 cancers-13-01106-f001:**
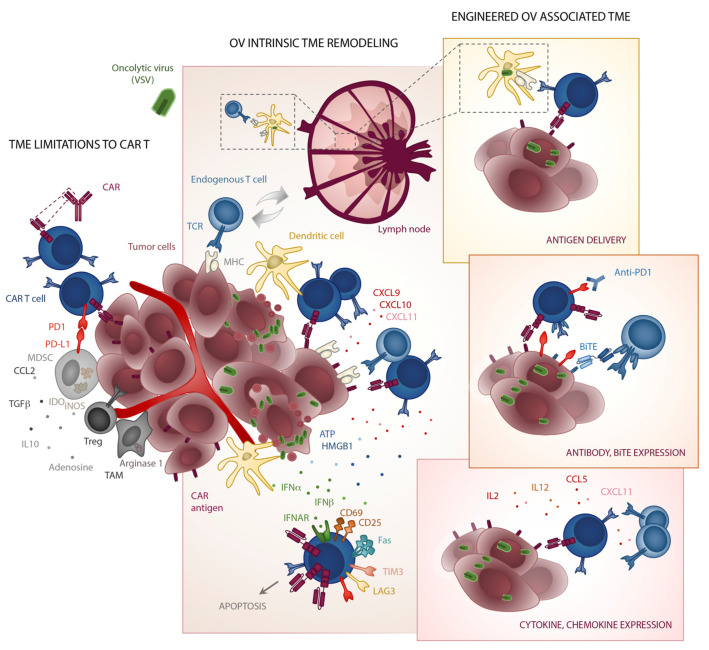
Strategic combination of oncolytic viruses with CAR T cells. The TME presents many immunosuppressive barriers to CAR T trafficking (high levels of CCL2, low levels of T cell chemotactic chemokines), as well as functionality through cytokines (TGFβ and IL10), metabolic dysregulation (arginase 1, inducible NO synthase (iNOS), indoleamine 2,3-dioxygenase (IDO), and CD39 and CD73 production of adenosine), and inhibitory ligands (PDL1 etc.). Many of these factors are expressed by tumor associated macrophages (TAMs), myeloid derived suppressor cells (MDSCs), regulatory T cells (Tregs) or the tumor cells themselves. Viral infection and oncolysis of tumor cells lead to the production of type I interferons (IFNs), danger-associated molecular pattern molecules such as HMGB1 and ATP, and CXCL9, 10, and 11 which in turn recruit additional T cells and dendritic cells. Exposure to high level type I IFN can also have inadvertent negative consequences for CAR T cells leading the upregulation of various inhibitory receptors including PD1, TIM-3 and LAG-3, as well Fas, leading to apoptosis. In contrast to some of these intrinsic properties, OVs can be armed with transgenes such as cytokines (IL2, IL12), chemokines (CCL5, CXCL11), checkpoint blocking antibodies (anti-PD1, etc.), BiTEs (EGFR, etc.) or the CAR antigen itself (CD19, etc.).

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
