# Peer review of "Parking CAR T Cells in Tumours: Oncolytic Viruses as Valets or Vandals?"

_cancers, 2021, doi:10.3390/cancers13051106_

Round 1

Reviewer 1 Report

Authors of the review provide a comprehensive introduction into both CAR-T cell technology as well as oncolytic viruses. 

As much as CAR-T cells have been successful in blood malignancies, solid tumours with their structure and highly immunosuppressive microenvironment, have posed an obstacle for successful implementation of adoptive T cell therapies in clinic.

The combination therapy strategies using oncolytic viruses and CAR-T cells may be a potential solution to this problem.

The topic of supplementation of CAR-T therapies with oncolytic viruses has been previously covered in multiple reviews.

The novelty of this review is concentrating both on the benefits that using oncolytic viruses brings, as well as on the less desired effects of their action and what can this potentially mean for the clinical outcome.

The review is clearly written and easy to follow. The authors have divided it into clear sections, with a comprehensive summary figure bringing the discussed knowledge together.

Comment 1: Minor typos: e.g line 18

Comment 2: TRAC modification with CAR insertion (line 69-70) is used in case of allogeneic CAR-T cells manufacturing in order to prevent GVHD as a side effect. Throughout the paragraph authors are discussing mostly autologous approach. The reviewer suggests to either clarify that in this case authors are discussing allogeneic CAR-T manufacturing strategies or mention the non-viral ways of CAR delivery. (e.g. transposon). Alternatively reviewer suggests introducing briefly allogeneic CAR-T cells in the general CAR-T paragraph.

Comment 3: Please introduce a legend to Figure 1.

Comment 4: Please keep the figure consistent with all mechanisms described in text (e.g. since TGF and IL10 are part of the figure so should remaining factors produced by immunosuppressive cells of TME: arginase 1, iNOS, IDO)

Similarly there is no TIGIT on CAR-T cells, whereas all other receptors discussed are part of the figure.

Comment 5: line 136 change TAMS to TAMs

Comment 6: Suggestion to change working from “beautifully engineered” to “meticulously engineered” CAR.

Comment 7: Authors use the word parking in many parts of the text. The reviewer suggests including a small section discussing if using oncolytic viruses can have impact of CAR-T cell trafficking through the tumor. Literature such as the paper mentioned below might be of relevance:

Combining Vascular Normalization with an Oncolytic Virus enhances Immunotherapy in a preclinical model of advanced stage ovarian cancer; Matuszewska et al., Clinical Cancer Research 2019.

Reviewer agrees that to a large extend this effect of OV is covered in “Viruses as micro-pharmacies for T cells section”

Reviewer 2 Report

Although the paper presents an interesting topic and the topic is written in an active voice with relatively obvious information, I have some concerns that need to be clarified and addressed.

Major concerns

  1. This topic has recently been covered by several review articles; therefore, it would be better to add information about the period covered by this paper. This is to ensure that the paper is up to date.
  2. Clear information about what is novel in this review compared with the recently published reviews is required. Therefore, it is significant to address this point in the text.
  3. Another section named, Methodology would be useful to be added to provide the readers with information about what type of strategies/methods were used or followed to ensure the quality of processing the collected data and the search outcomes. Also, information about the used databases for collecting and or extracting the data. Alternatively, this information could be highlighted in the Introduction section.
  4. The authors presented general information about oncolytic viruses without giving any information about what types of OV were used in the therapy or predicted to be involved. Please, address this point.
  5. Line 100. Combination OV and T cell Therapies: Driving CAR T to the Tumour. In this section, additional studies should be added to support the potential efficacy of combining both therapies (OV and T cell). In my opinion, it would be better to add more information about the possible side effects of both treatments and their limitations. Also, the opportunities of enrolling in a trial are needed to be discussed. These points might be discussed in a new section.   

Minor concerns

Figure 1. Please add a description to this figure. For non-specialized readers, it is hard to understand the content and the point that you want to report. 

Reviewer 3 Report

The work entitled “Parking CAR T Cells in Tumours: Oncolytic Viruses as Valets or Vandals?” submitted by Evgin & Vile is a particularly well-written and comprehensive review of transgenic T cell and oncolytic viral therapies. It describes the limitations and advances in each individual field and provides a thorough investigation of the possibilities and benefits of combining these therapies.

The review is well-crafted, cleverly constructed, and beautifully illustrated. There seems to be no obvious omission in its scope and this reviewer sees no reason why it shouldn’t be published in its current form. I think it will provide a very valuable overview of the field to readers of this journal.

Reviewer 4 Report

In this review, Evgin et al. summarize the key strengths and weakness of CAR-T cells and OV as monotherapies for cancer treatment and discuss the current strategies to combine these two modalities to overcome some of the barriers that both therapies find in solid tumors. The review is very well written and exhaustive. The authors have substantial preclinical and translational experience in oncolytic virotherapy and cell therapy, which allows them to make sound, experience-based recommendations and conclusions where appropriate. Here are some minor comments:

The statement in line 68: “…thus prompting use of non-viral means to introduce the CAR into genomic safe harbours such as the TRAC locus using CRISPR/Cas9[28, 29]” is not accurate in terms of “non-viral means” as the work referenced is using AAV6 vectors to deliver the CAR construct. Please, correct.

Reference 80, line 174: “Immune correlative studies in clinical studies have corroborated preclinical findings, showing that talimogene laherparepvec (HSV) and reovirus treatment promotes an increase CD8 T cell density in post treatment biopsies [78, 79], and facilitates T cell priming against tumour antigens [80]”. Reference #80 reports data on measles virus. As written this sentence is misleading.

The fact that viral antigens may be dominant over tumour neoantigens should be reviewed when discussing epitope spreading.

Reference #90 should be referenced in the text before ref. #89, as this was the first article to be published that combined CAR-T cells with OVs.

The following sentence is misleading: “In order to overcome antigen loss or heterogeneous expression, Bispecific T-cell engager (BiTE) technology has been used which links an anti-CD3 scFv to a tumour antigen specific scFv, and thus bypasses both the TCR and the CAR to engage effector function”. BiTEs are not designed to overcome antigen loss. The strategy described by Wing et al (combination of CAR-T, OV and BiTEs) is designed to overcome antigen loss. Also, the statement that BiTEs bypass the TCR is a bit confusing, as BiTEs activate T cells through the CD3, that is part of the TCR complex. It would be more accurate to say that BiTEs bypass the TCR-MHC interaction.

Round 2

Reviewer 2 Report

Dear Authors,

The paper has been improved, however, one point has not been addressed. I asked you to provide information about the period of studies covered by this paper (years from...to...) and the databases used to collect the acquired studies (for example, Web of Science, Scopus, Pubmed,....). Such a procedure is called a Methodology. Such information will help other researchers in the future to write another updated review on the same topic. By the way, it is important to provide information about the used databases since numerous journals are not indexed in the same databases, for example in Scopus and or Pubmed or even other online databases. Therefore, this point is important to ensure that you did not miss any available data.